# The Engineering, Expression, and Immobilization of Epimerases for *D*-allulose Production

**DOI:** 10.3390/ijms241612703

**Published:** 2023-08-11

**Authors:** Jin Hao Tan, Anqi Chen, Jiawu Bi, Yee Hwee Lim, Fong Tian Wong, Dave Siak-Wei Ow

**Affiliations:** 1Microbial Cell Bioprocessing, Bioprocessing Technology Institute, Agency for Science, Technology and Research (A*STAR), Singapore 138668, Singapore; tan_jin_hao@bti.a-star.edu.sg; 2Chemical Biotechnology and Biocatalysis, Institute of Sustainability for Chemicals, Energy and Environment, Agency for Science, Technology and Research (A*STAR), Singapore 138665, Singapore; chen_anqi@isce2.a-star.edu.sg (A.C.); wongft@imcb.a-star.edu.sg (F.T.W.); 3Molecular Engineering Lab, Institute of Molecular and Cell Biology, Agency for Science, Technology and Research (A*STAR), Singapore 138673, Singapore; bi_jiawu@imcb.a-star.edu.sg; 4Department of Food Science and Technology, National University of Singapore, 2 Science Drive 2, Singapore 117542, Singapore; 5Synthetic Biology Translational Research Program, Yong Loo Lin School of Medicine, National University of Singapore, 10 Medical Drive, Singapore 117597, Singapore

**Keywords:** *Bacillus subtilis*, biomanufacturing, *D*-allulose, *Escherichia coli*, metal nanoparticle, protein engineering, rare sugar, sweeteners

## Abstract

The rare sugar *D*-allulose is a potential replacement for sucrose with a wide range of health benefits. Conventional production involves the employment of the Izumoring strategy, which utilises *D*-allulose 3-epimerase (DAEase) or *D*-psicose 3-epimerase (DPEase) to convert *D*-fructose into *D*-allulose. Additionally, the process can also utilise *D*-tagatose 3-epimerase (DTEase). However, the process is not efficient due to the poor thermotolerance of the enzymes and low conversion rates between the sugars. This review describes three newly identified DAEases that possess desirable properties for the industrial-scale manufacturing of *D*-allulose. Other methods used to enhance process efficiency include the engineering of DAEases for improved thermotolerance or acid resistance, the utilization of *Bacillus subtilis* for the biosynthesis of *D*-allulose, and the immobilization of DAEases to enhance its activity, half-life, and stability. All these research advancements improve the yield of *D*-allulose, hence closing the gap between the small-scale production and industrial-scale manufacturing of *D*-allulose.

## 1. Introduction

*D*-allulose, also commonly known as *D*-psicose, is an epimer of *D*-fructose at the C3 position [1]. It is a rare sugar found in extremely low amounts in nature. With approximately 70% of sweetness but only 10% of calories and a 0.3% energy deposition of sucrose, *D*-allulose has been widely discussed as a suitable replacement for sucrose, and the United States Food and Drug Administration (FDA) classifies *D*-allulose as generally recognized as safe (GRAS). Amidst the escalating concerns surrounding the potential toxicities of artificial sweeteners, there is a notable shift towards embracing natural sugars as a viable alternative [2]. One study (*n* = 102,865) reported an increased cancer risk linked with the consumption of artificial sweeteners, notably aspartame and acesulfame-K, found in many food and beverage brands [3]. In comparison to artificial alternatives, past research has found that the naturally occurring *D*-allulose exhibits a multitude of health benefits, such as anti-hyperglycaemic and anti-hyperlipidaemic effects [4], anti-inflammatory capabilities [5], anti-oxidant properties [6,7], anti-obesity activity through the suppression of body fat deposition [8], and neuroprotection effects [9], to list just a few.

*D*-allulose production primarily involves the Izumoring strategy, which utilises the epimerase enzyme family for the bioproduction of all hexose sugars [10]. Specifically, these epimerases are typically of microbial origin, deriving from the ketose 3-epimerase family as *D*-tagatose 3-epimerase (DTEase), *D*-allulose 3-epimerase (DAEase) (also named *D*-psicose 3-epimerase, DPEase), all of which can isomerize *D*-fructose into *D*-allulose (Figure 1). It is worth noting that DTEase exhibits a preference for *D*-tagatose over *D*-fructose, unlike DAEase. Despite the high selectivity of ketose 3-epimerase, the equilibrium of this biotransformation disfavours *D*-allulose, generally in a <40% conversion. Additionally, there are certain limitations that further reduce the yield of *D*-allulose, such as the low thermal stability of DTEase and DAEase and non-enzymatic browning reaction of *D*-allulose [11]. While the typical optimal temperature of DTEase and DAEase falls between 40 and 70 °C, their relatively low thermostability causes the enzymes to lose their function quickly [12]. As a result, a constant supply of these enzymes is required for *D*-allulose production. The optimal pH condition of 7.5–9.0 for these enzymes, together with high reaction temperatures, causes the non-enzymatic browning of *D*-allulose, reducing the yield while further complicating the purification process. Low yields can also be attributed to limitations caused by thermodynamic equilibrium, resulting in a low conversion rate of *D*-fructose to *D*-allulose [12].

There are two general methods that have been established, namely, the use of immobilized enzymes and that of whole-cell biotransformation. The comprehensive reviews of biosynthesis and whole-cell biotransformation recently carried out by Chen et al. [5], Jiang et al. [13], and Xia et al. [14] will not be covered here. In this review, we will focus on certain innovative and novel engineering, expression, and immobilization methods of DAEase to improve the industrial manufacturing of *D*-allulose, focusing particularly on reports from the last two years.

## 2. Characterization of Newly Identified DAEases

Even though working with and improving on previously known DAEases is one of the approaches used to further the industrial manufacturing of *D*-allulose, identifying more variants of DAEase originating from different sources can widen our mechanistic understanding of these enzymes, as well as provide resources as new starting points for enzyme engineering towards industrial applications.

In the last two years, there have been at least three new notable identified DAEase that were reported and characterized (Table 1). The DAEase from *Arthrobacter psychrolactophilus* B7 (ApDAEase) [15], *Caballeronia insecticola* (CiDAEase) [16], and DPEase from *Iocasia fonsfrigidae* SP3-1 (IfDPEase) [17] were expressed in *Escherichia coli* BL21(DE3) and subsequently purified and characterized. 

Originally an uncharacterized putative protein (GenBank Accession: WP_110484351), the genetic sequence for ApDAEase was discovered from the whole-genome sequence of *A. psychrolactophilus* strain B7 (GenBank Accession: NZ_QJVC01000003.1). The eventual DNA sequence determined for ApDAEase is phylogenetically close to a stable and effective DAEase from *Arthrobacter globiformis* M30 [15]. The expressed ApDAEase displayed excellent thermostability, with a half-life (*t*_1/2_) of 128.4 min at its optimal temperature of 70 °C. At 55 °C and 65 °C, it was reported to have *t*_1/2_ of 1155.2 min and 223.6 min, respectively [15]. The addition of Mg^2+^ extends the *t*_1/2_ at 55 °C to 1386.3 min and increases the enzymatic activity by up to 500% under optimal conditions. ApDAEase also had the highest turnover number (*k_cat_*) compared to any other wild-type DAEase to date. The calculated rate of conversion of *D*-fructose to *D*-allulose was 27% [15]. 

In the search for new DAEase, the amino acid sequence of *Pseudomonas cichorii* DTEase was used as a template for the NCBI BLAST tool. Li et al. [16] discovered a putative xylose isomerase (XI) from *C. insecticola* (GenBank Accession: BAN26336.1) after various in silico analyses. The resultant expressed CiDAEase had a slightly better conversion rate of up to 31% at the optimal temperature of 65 °C with 500 g/L of *D*-fructose after 45 min, as compared to ApDAEase. While the addition of Mg^2+^ also improved activity, the optimal metal ion for CiDAEase was Mn^2+^. When used in a multienzyme cascade reaction, a conversion rate of more than 80% was achieved in 5 h [16]. 

A new addition to the repertoire is IfDPEase, derived from the anaerobic *Iocasia fonsfrigidae* sp. SP3-1. Isolated from a salt evaporation pond, this anaerobic bacterium exhibits exceptional halophilic characteristics, thriving in growth media containing 20% (*w*/*v*) NaCl. [17]. Along with this special behaviour, the low level of similarity between IfDPEase and other ketose 3-epimerases led Wulansari et al. to hypothesize that IfDPEase possesses new functions. IfDPEase displayed the highest activities at 50 °C, a much lower temperature than that of the others. It is also a metal-dependent enzyme, with Mn^2+^ as its optimal metal ion and a neutral pH of 7.5 [17]. IfDPEase was also discovered to be halophilic, maintaining consistent activity in the presence of NaCl up to 500 mM. The authors also tested IfDPEase activity in fresh coconut water due to its pH of 6.5 and the presence of manganese ions. Approximately 26.77% of the *D*-fructose in the coconut water was converted to *D*-allulose after 5 min of incubation at 50 °C. Under optimal conditions, IfDAEase converted 36.1% of *D*-fructose to *D*-allulose in 5 min [17]. 

ApDAEase’s high turnover number, its excellent thermostability, CiDAEase’s higher conversion rate potential, and the high yield of IfDPEase with short reaction times provide these epimerases with the potential to be used in *D*-allulose on an industrial scale. 

## 3. Engineering of DAEases

### 3.1. Engineering for Improved Thermotolerance

DAEase are naturally thermolabile epimerases, and this characteristic is one of the key bottlenecks limiting the application of the Izumoring strategy for the bioproduction of *D*-allulose [21]. In an industrial context, higher temperatures are commonly employed to accelerate reactions and improve production efficiencies. During the bioconversion of *D*-fructose into *D*-allulose, the rapid depletion of functional enzymes results in not only poor efficiencies but also virtually no recycling of the biocatalyst, driving production costs up. Thermostability is a hallmark of a good biocatalyst, allowing for greater operational stability and kinetic efficiency when operated in elevated temperature ranges [22] or longer lifetimes when operated at cooler temperatures. Consequently, a significant number of recent engineering endeavours involving approaches such as direct evolution, computational design, or fusion tags have been focused on enhancing the thermal stability of DAEase (Table 2, Figure 2).

The directed evolution of enzymes is a well-established strategy used to engineer enzymes with more desirable characteristics. However, without a robust high-throughput screening method, directed evolution may prove to be both labour- and capital-intensive [22]. Feng et al. were able to develop a novel high-throughput screening assay for evolving a DAEase from *Clostridium cellulolyticum* H10 (CcDAEase), which was reported to be thermotolerant with a *t*_1/2_ of 24 min at 55 °C [22]. An error-prone PCR process was used to generate the DAEase mutant library, and each mutant was overexpressed in *E*. *coli* BL21(DE3). The high-throughput selection method developed made use of a coupled enzyme assay involving XI and DAEase. *D*-fructose was equilibrated at 60 °C in DAEase, followed by heat inactivation and the addition of XI to convert any un-catalysed *D*-fructose into *D*-glucose. The resulting ketose content was assessed using a chromogenic assay, revealing promising mutants with 1.5- to 1.8-fold improved residual activity after heat treatment compared to the wild type. The improved mutants contained single mutations and were subsequently combined to produce a double mutation variant with a 2.4-fold residual activity improvement over the wild type [22]. Further evolution using this improved mutant gave rise to a new mutant with residual activity improved by up to 2.6-fold and an inactivation *t*_1/2_ improved by up to 9.5-fold at 60 °C relative to the wild type. This strategy offers an effective high-throughput screening method and potential targets for industrial production.

Computational tools empower enzyme engineers to perform better structural analysis and modelling to create better and more robust rational designs with which to improve the thermostability of DAEases, allowing for more avenues for solving industrial bottlenecks. Chen and co-workers engineered previously characterized DAEase derived from thermophilic *Thermoclostridium caenicola* and further improved its thermostability via computer-assisted targeted mutagenesis based on a *ΔΔG_fold_* calculation and ordered recombination mutagenesis [23]. Modelling based on structural homology aided in the selection of mutational hotspots, while the calculated free energy of folding (*ΔG_fold_*) values provided parameters for the in silico generation of mutants. This approach resulted in mutants with a *T_m_* of 79.48 °C, more than 12 °C higher than that of the wild type, which also had a higher *t*_1/2_ of more than 10 h at 65 °C and an improved *T_opt_* of 75 °C compared to the wild-type enzyme, which had a *t*_1/2_ of less than 5 min at the same temperature [23]. In another study, Wang et al. [25] also adopted a rational design strategy which incorporated proline residue substitutions and constructed double mutant designs through homology modelling and computational structural analysis. The resultant mutant was able to retain comparable enzymatic activity while showing a *t*_1/2_ increased up to 2-fold [25].

Among the various characterized DAEases, the enzyme isolated from *Agrobacterium* sp. ATCC 31,749 (AsDAEase) was reported to have a relatively high specific activity and yield in *E. coli* [28]. Although the protein could reach a high yield, it faced rapid heat inactivation when exposed to temperatures of or above 60 °C, typically within 30 min [24]. In an effort to improve its thermostability and solubility, Tseng et al. adopted the strategy of implementing a fusion tag derived from the C-terminal acidic tail of α-synuclein (ATS), which is both thermostable and possesses chaperone activity. In Tseng’s study, a 22-amino-acid-long ATS peptide sequence (residue 119–140) was fused to both wild-type and thermostable AsDAEase mutants before expression in *E. coli* BL21(DE3) [24]. A total of four different combinations of enzymes were characterized: the wild type, wild type fused with ATS, mutant (I33L/S213C), and mutant fused with ATS (I33L/S213C-ATS fusion, known as LCATS). Compared to the wild type, the LCATS demonstrated a significantly improved thermostability up to 65 °C, along with 19-fold increase in its half-life. Its catalytic efficiency was also slightly higher than that of the wild type at 64.1 ± 12.0 compared to 41.2 ± 12.0 min^−1^ mM^−1^ [24]. 

### 3.2. Engineering for Improved Acid Resistance

It is known that the optimal pH range for native DAEases is within 7.5–9.0 [12]. This inherent characteristic of DAEases makes them unsuitable for direct *D*-allulose synthesis in low-pH environments, such as acidic fruit juices. Yet, non-enzymatic browning reactions frequently result in by-product formation under high-temperature, alkaline conditions, hindering the industrial viability of DAEase in its native optimal condition [26]. A computational analysis of CcDAEase and a DAEase derived from *Dorea* sp. CAG317 with a reported optimal pH of 6.0 revealed a correlation between an increased number of negatively charged residues and a drop in the optimal pH of the enzyme. The mutation of non-conserved residues into negatively charged ones produced mutants with up to 26% higher residual activity when compared to the wild type at pH 5.0. Combining a semirational design with random mutagenesis and high-throughput screening enabled further functional improvements of the CcDAEase design. However, while the mutants had higher acidic tolerance, for most of them, this came at the cost of their thermostability. Two out of the three mutants showed decreased thermostability at 55 °C, and the remaining mutant (M3) showed similar thermostability to the wild type. The optimal pH for M3 is 7.5, though it can retain activity (approximately 10%) at a pH as low as 3.0. Li et al. tested the mutant M3 in a multi-enzyme cascade reaction in acidic juices such as kiwi juice, mango juice, grape juice, and orange juice to produce *D*-allulose. The proportion of *D*-allulose produced as a percentage of the total sugars reached 16.3% (13.7 g/L), 14.8% (14.9 g/L), 17.1% (15.40 g/L), and 16.1% (7.80 g/L), respectively, for the above juices [26]. Compared to the wild type, which could only achieve 3.85 g/L of *D*-allulose sugar in grape juice, acidic tolerance engineering provides a promising strategy for developing next-generation enzymes for industrial production purposes.

Through different engineering strategies, DAEase enzymes could be improved, and their thermostability and environmental tolerance could be enhanced (Table 2). From directed evolution supported by high-throughput screening to computer-assisted rational design and even the introduction of peptide fusion tags, recent studies were able to demonstrate the promising capacity of DAEase enzymes to be developed into more thermostable and resilient biocatalysts suitable for industrial production.

## 4. Utilization and Engineering of *B. subtilis* for *D*-Allulose Biosynthesis

In addition to *E. coli*, the endotoxin-free GRAS *Bacillus subtilis* expression system is often used to express DAEase. In addition, *B. subtilis* works as a whole-cell biocatalyst to produce *D*-allulose. This reduces the need to purify DAEase for *D*-allulose production. However, *B. subtilis* consumes *D*-fructose as a source of carbon for growth, reducing the yield of *D*-allulose. In order to enhance the production of this rare sugar through whole-cell biotransformation, Zhang et al. [29] employed *B. subtilis* WB600 as an expression host for *D*-allulose 3-epimerase from *Sinorhizobium fredii* (SfDAEase). Additionally, they conducted a knockout of three genes to restrict the carbon flow of *D*-fructose towards the Embden–Meyerhof–Parnas pathway and the Krebs cycle. This allows for most of the *D*-fructose absorbed by *B. subtilis* to be converted into *D*-allulose when SfDAEase is expressed in the cell. To enhance *D*-allulose production, the authors further conducted a series of engineering steps, including screening, fine-tuning, and the development of a dual promoter variant with a 6.2-fold increase in the mRNA transcription level. This optimization led to impressive results in an optimized 5 L fed-batch bioreactor, with a significant 74.2 g/L *D*-allulose achieved after 64 h. Importantly, the high yield of 0.93 g *D*-allulose/g *D*-fructose surpassed the previous yield of 0.13 g/g, making it the highest *D*-allulose yield reported using *B. subtilis* fermentation to date [29]. 

In another study, CcDAEase expressed in *B. subtilis* was enhanced through a similar promoter-screening strategy. After obtaining mutants of CcDAEase with improved specific activity and thermal stability through protein engineering in *E. coli* BL21, Liu et al. [21] transformed the most successful variant into *B. subtilis*. In their fermentation process, glycerol is added as a cost-efficient carbon source. Nevertheless, due to carbon catabolite repression, the total enzyme activities of CcDAEase exhibited a notable decrease from 80 U/mL when the medium contained 5 g/L of glycerol to 6.74 U/mL when the glycerol concentration was increased to 30 g/L. As noted by Liu et al., this observation confirms the presence of a catabolite-responsive element (CRE) box upstream of the promoter region, a potential challenge for industrialization. With reference to Weickert and Chambliss [30], Liu et al. achieved a *B. subtilis* mutant through CRE box engineering with 282.43% relative enzyme activity, as compared to the wild-type CcDAEase. Finally, through high-density fermentation in a 3 L bioreactor, they managed to achieve 4971.50 U/mL enzyme activity in 68 h. This represents a notable improvement in both enzyme activity and the titer compared to a previous study where CcDAEase was employed to produce *D*-allulose, achieving an enzyme activity of 2246.00 U/mL in 88 h [31].

Generally, the optimal pH of most DAEases lies in the neutral to alkaline range. The production of *D*-allulose in high-temperature and -pH conditions speeds up the non-enzymatic browning of the rare sugars, reducing the yield while complicating the downstream processes. Producing *D*-allulose in acidic conditions can lessen these complications. To this end, Hu et al. [11] recently achieved efficient *D*-allulose synthesis under acidic conditions by expressing DsDAEase and CcDAEase in tandem in *B. subtilis* 168. The two DAEases had complimentary properties. The activity of DsDAEase peaked around 60 °C and reduced to below 80% relative activity at higher temperatures, while CcDAEase maintained its activity above 80% within the 60–75 °C range, but DsDAEase was less affected by changes in pH as compared to CcDAEase. Furthermore, both epimerases displayed the highest activity in the presence of Co^2+^. The catalytic activity of the strain with DsDAEase, followed by CcDAEase, displayed the highest enzymatic activity in acidic conditions, with 18.9 U/mL at pH 6.5 compared to the single-enzyme strains of DsDAEase and CcDAEase, with 17.2 U/mL and 16.7 U/mL, respectively. However, these activities are still relatively low compared to the activities of the same enzymes under alkaline conditions. Hu et al. [11] improved the expression level of the DAEases by introducing auto-inducible promoters in front of the DAEase genes. Four dual-promoter strains were constructed, with the best strain achieving 154.4 U/mL at pH 6.5. With this strain as a template, two other promoters were tested. The final engineered strain, *Bs168*/pMA5-P_spoVG_-*DS*dpe-P_srfA_-*RC*dpe, had a specific activity of 228.5 U/mL at pH 6.5; this is a >12.0-fold improvement in enzyme activity compared to the individual DAEases at the same pH. With a bioreactor runtime of 30 h, an enzyme activity of 387.7 U/mL was achieved. At 42 h, the authors reported the highest enzyme activity of 480.1 U/mL [11].

With multiple different approaches, the researchers managed to significantly improve the performance of the *B. subtilis* whole-cell biosynthesis of *D*-allulose. It is likely that using a combination of these approaches will yield a strain with an even greater performance and efficiency to produce *D*-allulose on an industrial scale with lower costs and shorter batch times.

## 5. Immobilization

Various immobilization techniques have been explored by researchers to overcome the limitations associated with free allulose epimerases for *D*-allulose production. These techniques aim to enhance the stability and longevity of the epimerases, enable their reuse, and facilitate their effective separation from the reaction mixture, thereby improving the industrial application of free allulose epimerases from both cost and operational perspectives.

An early example of epimerase immobilization was that of a DTEase from *Pseudomonas cichorii* ST-24 [32]. The DTEase expressed in *E. coli* JM105 was immobilized on Chitopearl beads, BCW 2510. A jacked column reactor of 200 g supporting DTEase (70 U/g resin) was operated continuously for 60 days at pH 7.0 and 45 °C. Despite a lower conversion of 25%, 20 kg of pure allulose was produced after the removal of the fructose in the mixture using Baker’s yeast. This example demonstrated the attractiveness of epimerase immobilization and its potential application for the large-scale production of allulose. Following this successful case, the immobilization of epimerases from various sources on different supports has been reported and reviewed [13,14,33]. 

Ideally, the immobilized epimerases should show retained or enhanced activity, with a significantly improved stability and half-life. The methods used for epimerase immobilization are generally physical adsorption, encapsulation, ionic binding, or covalent linkage onto a suitable support (Figure 3). The characteristics of the support material, such as a high affinity for protein, mechanical stability, rigidity, and reusability, should be taken into consideration. These immobilization methods exploit the effective attachment of non-active amino acid residues in the enzyme to the support without compromising the performance of the enzyme. While the selection of a suitable support is of great importance for epimerase immobilization, other factors such as enzyme loading, pH, temperature, metal ion, and leaching should also be taken into consideration to achieve an optimal performance of the immobilized enzyme. This section will provide a summary of the more recent literature on epimerase immobilization that was not covered in previous reviews (Table 3).

Metal nanoparticles are often explored as supports for enzyme immobilization due to their ability to be functionalized and large surface area. In this regard, Dedania et al. [34] studied titanium dioxide nanoparticles for the immobilization of DPEase from *Agrobacterium tumefaciens*, using glutaraldehyde as a cross-linking agent. The TiO_2_-immobilized DPEase was characterized using a suite of spectroscopic techniques. The immobilized DPEase showed a *t*_1/2_ of 180 min at 60 °C, which was higher than that of the free enzyme (3.99 min at 50 °C). The immobilized enzyme displayed a high conversion of fructose at 36:64 at equilibrium under the optimal conditions (pH 6.0, 60 °C, and in the presence of Mn^2+^), but its activity declined rapidly after reuse, retaining only 50% and 20% of its initial activity after five and eight cycles, respectively. 

Recognizing the cobalt dependence of certain epimerases, ZIF-67, a Co^2+^-containing zeolitic imidazolate framework [38], has also been explored for the immobilization of epimerases. Xu et al. [35] prepared ZIF-67 on magnetic nanoparticles (ZIF-67@Fe_3_O_4_) and used these particles to support DAEase from *Agrobacterium tumefaciens.* Under optimal conditions (pH 8.0 and 55 °C), the immobilized DAEases showed a catalytic activity of 65.1 U mg^−1^ and a 38.1% conversion of fructose, both higher than the values for the free enzyme. The immobilized DAEase also displayed better thermal stability than the free enzyme over a range of temperatures (45–70 °C) and maintained > 45% of its initial activity after eight cycles. In a related study, Yang et al. [36] identified a novel DAEase from *Ruminiclostridium papyrosolvens* and briefly investigated its immobilization onto ZIF-67 nanoparticles. Although the immobilized enzyme showed activity similar to the free enzyme under optimal conditions (pH 7.5 and 60 °C), it had better tolerance to higher temperatures (65–80 °C) and pH (pH 8–10). The immobilized enzyme activity retained 56% of its initial activity after five cycles.

As the conventional immobilization of epimerases results in a random orientation of adsorbed proteins on the surface of the support, which can lead to poorer activity due to active site pockets that are inaccessible to the substrate, Gao et al. [37] explored the potential benefit of a directional immobilization approach to minimise the effect of the steric hindrance of the enzyme active site on the substrate. In this approach, CcDAEase was directionally immobilized onto a SpyCatcher-SpyTag-modified [39] epoxy resin, ES-103B, and compared to the randomly immobilized enzyme. Although the directionally immobilized DAEase exhibited similar thermal and pH stability compared to the randomly immobilized DAEase, it displayed superior catalytic efficiency with a 2.4-fold increase in *k_cat_/K_m_*. Whilst both the directionally and randomly immobilized DAEases showed similar recyclability (>75% relative activity after 7 cycles) and storage stability (>80% relative activity after 25 days), the directionally immobilized DAEases produced 30% more allulose than the randomly immobilized DAEases when applied to the production of a mixed fruit juice containing *D*-allulose.

In summary, the immobilization of allulose epimerases has been investigated as a means to improve their stability, reusability, and catalytic performance in the production of *D*-allulose. Despite the progress in this regard, the overall performance of the immobilized epimerases, especially their half-lives, recyclability, ease of operation, and cost-effectiveness, remain to be further improved for industrial application. As such, further efforts are required to develop better and more practical immobilization techniques for allulose epimerases.

## 6. Summary and Future Vision

In this review, we discussed how new epimerases can be discovered [15,16,17], as well as methods to improve their stability and activity via protein engineering [21,22,24,25,28], the application of *B. subtilis* [11,25,29], and the potential of immobilization techniques to enhance performance [34,35,36,37]. These advancements have the potential to significantly reduce production time and costs associated with *D*-allulose biosynthesis. With increasing societal awareness of the health risks associated with excessive sugar consumption, the demand for safer sugar alternatives with low or zero calorie contents continues to surge. Building upon this success, we can further push the boundaries of engineering and innovation using emerging technologies such as artificial intelligence (AI) generative systems, digital twins, and more. In recent years, much research has been directed towards harnessing the power of AI with respect to protein engineering. Machine learning models such as the transformer protein language model, as well as AI-based protein structure prediction, are promising for the identification of suitable sequences for protein engineering [40,41]. Also, several newly generated protein sequences developed using a generative adversarial network were soluble and functional [42]. By harnessing these advancements, we can develop sustainable manufacturing processes that efficiently meet the growing demand for allulose in a cost-effective manner.

## Figures and Tables

**Figure 1 ijms-24-12703-f001:**
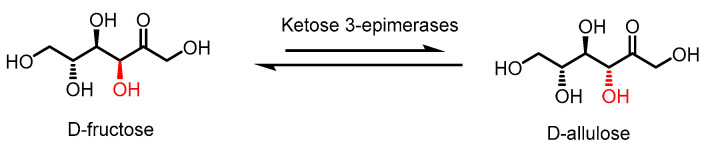
Epimerization of *D*-fructose by ketose 3-epimerases to produce *D*-allulose.

**Figure 2 ijms-24-12703-f002:**
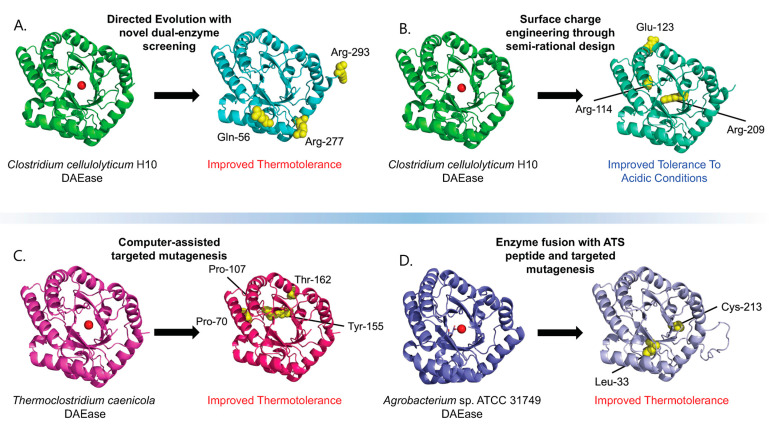
Outcomes of the various DAEase engineering studies. The red circle on the original un-modified proteins represents the substrate and showcases the enzyme binding pocket. Mutated residues are displayed as yellow spheres and labelled. Protein structures of engineered mutants were predicted via AlphaFold [27]. (**A**) Through directed evolution made possible using a novel, high-throughput, dual-enzyme screening system, the wildtype CcDAEase was engineered into a more thermotolerant mutant enzyme [22]. (**B**) The wild-type CcDAEase was engineered via a semi-rational design, and the surface charge engineering produced a mutant enzyme capable of tolerating lower pH environments [26]. (**C**) DAEase from *Thermoclostridium caenicola* was engineered through computer-assisted mutagenesis to produce a mutant with higher thermotolerance [23]. (**D**) Fusion with a peptide derived from the acidic tail of alpha-synuclein and directed mutagenesis produced a mutant with improved thermotolerance [24].

**Figure 3 ijms-24-12703-f003:**
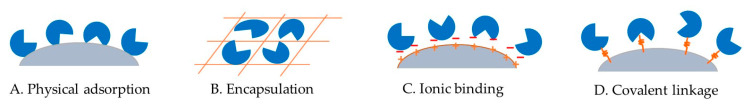
Illustration of ketose 3-epimerase immobilization via (**A**) physical adsorption, (**B**) encapsulation, (**C**) ionic binding, and (**D**) covalent linkage onto a solid support.

**Table 1 ijms-24-12703-t001:** Characteristics of expressed heterologous DAEases.

Enzyme Source	Expression Strain	Optimal Reaction Conditions		Metal Ion	Conversion (%)	*k_cat_* (s^−1^)	*K_m_* (mM)	*k_cat_/K_m_* (mM^−1^ s^−1^)	*t*_1/2_ (mins)	Reference
		pH	Temperature (°C)							
DAEase from *Anthrobacter psychrolactophilus*	*E. coli* BL21(DE3)	8.5	70	Mg^2+^	27%	2920	738.7	3.953	1386.3 (55 °C + Mg^2+^) 128.4 (70 °C)	[15]
DAEase from *Caballeronia insecticola*	*E. coli* BL21(DE3)	9	65	Mn^2+^	31%	204.05	137.7	1.482	Not reported	[16]
DPEase from *Iocasia fonsfrigidae*	*E. coli* BL21(DE3)	7.5	50	Mn^2+^	36%	12.82	21.31	0.602	Not reported	[17]
DAEase from *Bacillus* sp. KCTC 13,219 (DaeB)	*B. subtilis* RIK 1285	8	55	Mn^2+^	28%	367	130.6	2.810	36,000 (50 °C + Mn^2+^) 1320 (55 °C + Mn^2+^)	[18]
DAEase from *Clostridium cellulolyticum* H10	*E. coli* BL21(DE3)	8	55	Co^2+^	32%	54.05	17.4	3.106	570 (55 °C + Co^2+^)	[19]
DAEase from *Dorea* sp. CAG317	*E. coli* BL21(DE3)	6	70	Co^2^	30%	507.4	153	3.316	36 (60 °C)	[20]

**Table 2 ijms-24-12703-t002:** Engineering strategies employed to improve enzyme functionalities.

Enzyme Source	Expression Strain	Engineering Strategy	Functional Improvements	Reference
DAEase from *Clostridium cellulolyticum* H10	*E. coli* BL21(DE3)	Directed evolution supported by a novel dual-enzyme screening system	Improved thermostability	[22]
DAEase from *Thermoclostridium caenicola*	*E. coli* BL21(DE3)	Computer-assisted targeted mutagenesis	Improved thermostability	[23]
DAEase from *Agrobacterium* sp.	*E. coli* BL21(DE3)	Enzyme fusion with peptide derived from the acidic tail of alpha-synuclein (ATS)	Improved thermostability	[24]
DAEase from *Clostridium bolteae*	*E. coli* BL21(DE3)	Proline residue substitution to reduce unfolded enzyme entropy	Improved thermostability	[25]
DAEase from *Clostridium cellulolyticum* H10	*E. coli* BL21(DE3)	Directed evolution, combinatorial mutagenesis	Improved enzyme specificity, improved thermotolerance	[21]
DAEase from *Clostridium cellulolyticum* H10	*E. coli* BL21(DE3)	Protein surface charge engineering through semirational design and mutagenic library	Improved tolerance to acidic conditions	[26]

**Table 3 ijms-24-12703-t003:** Immobilization of epimerases for the production of *D*-allulose.

Enzyme Source	Support Material	Immobilization Method	pH	Temperature (°C)	Metal Ion	Reference
DPEase from *Agrobacterium tumefaciens*	Glutaraldehyde functionalised TiO_2_	D	6	60	Mn^2+^	[34]
DAEase from *Agrobacterium tumefaciens*	Cobalt-based magnetic MOF ZIF-67@Fe_3_O_4_	B	8	50	Co^2+^ (in ZIF-67)	[35]
DAEase from *Ruminiclostridium papyrosolvens*	ZiF-67	B	7.5	60	Co^2+^ (in ZIF-67)	[36]
DAEase from *Clostridium cellulolyticum*	Epoxy resin ES-103B with or without SpyCatcher-SpyTag	D	8	55	-	[37]

## Data Availability

No new data were created or analysed in this study. Data sharing is not applicable to this article.

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
