# Peer review of "The Engineering, Expression, and Immobilization of Epimerases for D-allulose Production"

_ijms, 2023, doi:10.3390/ijms241612703_

Round 1

Reviewer 1 Report

The article « Engineering, Expression, and Immobilization of Epimerases for D-allulose Production» is attracted to a relevant topic and has a high applied value.

The review covers fundamental works in this direction and modern literary sources.

It's known, that the sugar D-allulose is a potential replacement for sucrose with a wide range of health benefits. However, the process of D-allulose receiving is not efficient due to poor thermotolerance of the epimerases – enzymes which necessary to obtain it. This review described three newly identified enzymes DAEases that possess desirable properties for industrial-scale manufacturing of D-allulose. Hence closing the gap between the small-scale production and industrial-scale manufacturing of D-allulose.

The article is well structured, written in sufficient detail and logically.

However, there are some minor comments on the design of the manuscript:

1. Table 1 should be placed entirely on one page.

2. Caption to Table 2 and Table 2 itself should be placed on one page.

3. Section heading 3.2. and Section 3.2 itself should be on one page.

Besides, the conclusion should be expanded by adding materials on the prospects and possible current directions of research in the field of enzymatic production of D-allulose.

Author Response

The review covers fundamental works in this direction and modern literary sources.

It's known, that the sugar D-allulose is a potential replacement for sucrose with a wide range of health benefits. However, the process of D-allulose receiving is not efficient due to poor thermotolerance of the epimerases – enzymes which necessary to obtain it. This review described three newly identified enzymes DAEases that possess desirable properties for industrial-scale manufacturing of D-allulose. Hence closing the gap between the small-scale production and industrial-scale manufacturing of D-allulose.

The article is well structured, written in sufficient detail and logically.

  • We thank the reviewer for the positive feedback.

However, there are some minor comments on the design of the manuscript:

  1. Table 1 should be placed entirely on one page.
  • This has been addressed on the revised manuscript on page 3 line 80.
  1. Caption to Table 2 and Table 2 itself should be placed on one page.
  • This has been addressed on the revised manuscript on page 5 line 126.
  1. Section heading 3.2. and Section 3.2 itself should be on one page.
  • This has been addressed on the revised manuscript on page 7 line 202.
  1. Besides, the conclusion should be expanded by adding materials on the prospects and possible current directions of research in the field of enzymatic production of D-allulose.
  • We have expanded the discussion to include the following materials: “In recent years, much research has been put into harnessing the power of AI with respect to protein engineering. Machine learning models such as transformer protein language model, as well as AI-based protein structure prediction has been promising in identifying suitable sequences for protein engineering [40, 41]. Also, several newly generated protein sequences developed by a generative adversarial network were soluble and functional [42].” (page 11 line 383 - 388)

Reviewer 2 Report

In the review article “Engineering, Expression, and Immobilization of Epimerases for D-allulose Production”, Tan et al shortly summarized the recent findings for D-allulose production with epimerase processes. Three newly identified epimerases were introduced. The authors also summarized engineering strategies for obtaining evolved enzymes, such as directed evolution, computer-assisted targeted mutagenesis, enzyme fusion, proline residue substitution, and protein surface charge engineering through semirational design and mutagenic library. These engineered enzymes were used for improved thermotolerance and acid resistance. Some immobilization methods were also mentioned to overcome the limitations associated with free allulose epimerases for D-allulose production. This reviewer needs more discussions how the epimerases (processes) can be improved in the future. If available, the structural information of these engineered enzymes should be provided for the readers.

Author Response

In the review article “Engineering, Expression, and Immobilization of Epimerases for D-allulose Production”, Tan et al shortly summarized the recent findings for D-allulose production with epimerase processes. Three newly identified epimerases were introduced. The authors also summarized engineering strategies for obtaining evolved enzymes, such as directed evolution, computer-assisted targeted mutagenesis, enzyme fusion, proline residue substitution, and protein surface charge engineering through semirational design and mutagenic library. These engineered enzymes were used for improved thermotolerance and acid resistance. Some immobilization methods were also mentioned to overcome the limitations associated with free allulose epimerases for D-allulose production.

  • We would like to thank for the summary.
  1. This reviewer needs more discussions how the epimerases (processes) can be improved in the future.
  • We have expanded the discussion to include the following materials: “In recent years, much research has been put into harnessing the power of AI with respect to protein engineering. Machine learning models such as transformer protein language model, as well as AI-based protein structure prediction has been promising in identifying suitable sequences for protein engineering [40, 41]. Also, several newly generated protein sequences developed by a generative adversarial network were soluble and functional [42].” (page 11 line 383 - 388).
  1. If available, the structural information of these engineered enzymes should be provided for the readers.
  • We have added in Figure 2 (Outcomes of the various DAEase engineering studies.) on page 6 line 128.

Reviewer 3 Report

The review is devoted to the topical problem associated with the possible use of epimerases to obtain a potentially valuable product for the food industry D-allulose. The review is written in a good literary language, it analyzes numerous works on the engineering, expression, and immobilization of epimerases for the production of D-allulose. Significantly enough, this review focuses on DAEase to improve the industrial manufacture of D-allulose studied from the last two years. However, in this case, the reader does not form a comprehensive knowledge about these enzymes, for example; only three enzymes are considered in Table 3. Perhaps, new enzymes should be compared in this table with the most active (and/or thermostable) previously studied enzymes in order to visually (in the table) show the role of new DAEases on the panel of existing ones. For example, this review should include information on the new d-Allulose 3-epimerase of Bacillus sp KCTC 13219 and its expression in Bacillus subtilis cells (DOI: 10.1186/s12934-021-01550-1) and a number of others articles.

The review contains three tables and only one figure representing the reaction scheme. In my opinion, this is not enough for a review. For a more visual perception of information, it is required to illustrate the stated knowledge with additional figures. For example, the previous review of DAEases referenced by the authors in the Introduction section (doi: 10.3390/foods10092186 ) contains figures that contribute to a general understanding of the role and functioning of DAEases in the relevant sections. Current review, for example, in section 3. Engineering of DAEases, would greatly benefit if the authors made a figure-flowchart illustrating Engineering strategies employed to improve the enzyme. I also suggest a graphic illustration for section 5. Immobilization. And despite the fact that the review is written quite qualitatively, I think that it is necessary to make it a Major revision in order to think over and prepare the figures with high quality.

Author Response

The review is devoted to the topical problem associated with the possible use of epimerases to obtain a potentially valuable product for the food industry D-allulose. The review is written in a good literary language, it analyzes numerous works on the engineering, expression, and immobilization of epimerases for the production of D-allulose. Significantly enough, this review focuses on DAEase to improve the industrial manufacture of D-allulose studied from the last two years.

  • We thank the reviewer for the positive feedback.

However, in this case, the reader does not form a comprehensive knowledge about these enzymes, for example; only three enzymes are considered in Table 3. Perhaps, new enzymes should be compared in this table with the most active (and/or thermostable) previously studied enzymes in order to visually (in the table) show the role of new DAEases on the panel of existing ones. For example, this review should include information on the new d-Allulose 3-epimerase of Bacillus sp KCTC 13219 and its expression in Bacillus subtilis cells (DOI: 10.1186/s12934-021-01550-1) and a number of others articles.

  • We have added in 3 previously characterized enzymes on page 3, references 18-20.

The review contains three tables and only one figure representing the reaction scheme. In my opinion, this is not enough for a review. For a more visual perception of information, it is required to illustrate the stated knowledge with additional figures. For example, the previous review of DAEases referenced by the authors in the Introduction section (doi: 10.3390/foods10092186 ) contains figures that contribute to a general understanding of the role and functioning of DAEases in the relevant sections. Current review, for example, in section 3. Engineering of DAEases, would greatly benefit if the authors made a figure-flowchart illustrating Engineering strategies employed to improve the enzyme. I also suggest a graphic illustration for section 5. Immobilization. And despite the fact that the review is written quite qualitatively, I think that it is necessary to make it a Major revision in order to think over and prepare the figures with high quality.

  • Figure 2 (on Engineering of DAEase) and Figure 3 (on Immobilization) have been added on page 6 line 128 and page 10 line 323 respectively.

Round 2

Reviewer 3 Report

The authors took into account all the comments made, added figures to illustrate sections 3 and 5; expanded Table 1. The number of references in the original version was 35, which is somewhat small for a review. The resubmitted version contains 42 references.

As a result, the work looks much more coherent and easier to read. I consider that this manuscript can be accepted in its current form for publication.